# Effect of In Situ Heating on the Growth and Electrochromic Properties of Tungsten Trioxide Thin Films

**DOI:** 10.3390/ma17102214

**Published:** 2024-05-08

**Authors:** Jinfeng Xu, Xirui Li, Yong Zhang, Xueru Zhang, Jiaqin Liu, Yucheng Wu

**Affiliations:** 1School of Materials Science and Engineering, Hefei University of Technology, Hefei 230009, China; xujinfeng1031@163.com; 2Key Laboratory of Advanced Functional Materials and Devices of Anhui Province, Hefei 230009, China; jqliu@hfut.edu.cn; 3Anhui Jincen Composites Co., Ltd., Hefei 230009, China; lxr@jincen-cn.com; 4Institute of Industry & Equipment Technology, Hefei University of Technology, Hefei 230009, China; 5Instrumental Analysis Center, Hefei University of Technology, Hefei 230009, China; xueruzhang@hfut.edu.cn

**Keywords:** electrochromic, magnetron sputtering, tungsten trioxide, thin film, in situ heating, color analysis

## Abstract

Electrochromism has emerged as a pivotal technology in the pursuit of energy efficiency and environmental sustainability, spurring significant research efforts aimed at the creation of advanced electrochromic devices. Most electrochromic materials are used for smart window applications. However, current electrochromic materials have been applied to new energy vehicles, cell phone back covers, AR glasses, and so on. More application scenarios put forward more requirements for the color of the colored states. Choosing the right color change in the application will be the trend in the future. In this work, tungsten trioxide (WO_3_) thin films were prepared by adjusting the in situ heating temperature. WO_3_ with a crystalline structure showed excellent cyclic stability (5000 cycles), electrochromic performance (ΔT = 77.7% at 633 nm, CE = 37.1 cm^2^/C), relatively fast bleaching/coloring speed (20.0 s/19.4 s), and the darkest coloring effect (L* = 29.32, a* = 7.41, b* = −22.12 for the colored state). These findings offer valuable insights into the manipulation of smart materials and devices, contributing to the advancement of electrochromic technology.

## 1. Introduction

Electrochromism refers to the phenomenon that under the action of an electric field, the optical properties (absorption, transmission, reflection, etc.) of a material undergo a stable and reversible change, which is macroscopically manifested as a change in the color and transparency of the material. With the development of social science and technology, electrochromic materials were first used in smart windows to reduce a building’s energy consumption [1]. With the recent development of electronic products, intelligence, and wearability, electrochromic materials have been applied to new energy vehicles, cell phone back covers, AR glasses, and so on. It requires electrochromic films to have a high optical modulation range and cycling stability [2,3]. 

Tungsten trioxide (WO_3_), as a kind of classic electrochromic material, has a wide range of optical modulation, high cycling stability, and even energy storage properties [2,4]. The reaction Equation (1) of WO_3_ in cationic electrolyte is as follows:(1)$${WO}_{3}\left(bleached\right)+xM^{+}+{xe}^{-}⇌M_{x}{WO}_{3}\left(colored\right)$$

In the electrochromic reaction, Li ions and electrons are inserted into the WO_3_ lattice to form tungsten bronze materials, resulting in a coloring phenomenon. During the bleaching process, the Li ions and electrons are extracted from the WO_3_ lattice, restoring transparency. WO_3_ can be structurally divided into amorphous WO_3_ and crystalline WO_3_ [5]. In amorphous WO_3_, Li ions and electrons embedded in the amorphous WO_3_ can be removed more quickly, but its electrochromic performance declines rapidly and suffers poor durability as well. In crystalline WO_3_, the Li ions embedded in the color change process will be limited by the crystal structure, resulting in a lower response rate of color change. Potential traps may be formed during the embedding of ions and electrons [6]. But compared with amorphous WO_3_, its crystalline state promises excellent cycling stability [7]. In response to the application of electrochromic technology, researchers have focused on performance improvement in the past and now need to consider matching color transformation and application scenarios. Benedict Wen-Cheun Au et al. investigated the effect of different post-annealing temperatures on sol–gel spin-coated WO_3_ films [6]. K. Naveen Kumar et al. studied the effect of in situ heating and post-annealing processes on sputtered WO_3_ films [8]. Mengying Wang et al. focused on the chromaticity of complementary NiO//WO_3_ electrochromic devices by magnetron sputtering [9]. Color-changing devices by magnetron sputtering were focused on the chromaticity change. In this paper, WO_3_ thin films were prepared by magnetron sputtering, and the crystal structure of WO_3_ was changed by regulating the in situ heating treatment. We also implemented a chromaticity measurement to discern the subtle color differences under various electrochemical biases which may explore various potential of the film for practical applications. 

## 2. Materials and Methods

### 2.1. Preparation of ITO-Coated Conductive Glass

The 40 × 20 × 1.1 mm^3^ ITO-coated conductive glass was placed in acetone, ethyl alcohol, and ultrapure water, respectively, and ultrasonically cleaned for 15 min, followed by drying the ITO-coated conductive glass. The dried ITO-coated conductive glass was placed in a UV light cleaner (SHANGHAI ZHONGBIN TECHNOLOGY Co., Ltd., Shanghai, China) cleaned by ozone for 10 min to be used as a sputtering substrate.

### 2.2. Synthesis of WO_3_ Thin Films by Magnetron Sputtering

As shown in Figure 1, a piece of 25 × 20 × 1.1 mm^3^ glass was placed horizontally on another piece of 40 × 20 × 1.1 mm^3^ ITO-coated conductive glass (prepared in 2.1) to form an inverted T-shape to control the film area of 20 × 20 mm^2^. The tungsten target (99.99%) was used as the sputtering target. The pre-vacuum of atmosphere pressure in the vacuum chamber was controlled below 2 × 10^−4^ Pa. A control group without in situ heating was set up, which was noted as RT (substrate temperature of 20 °C), and different in situ heating temperatures (substrate temperatures) were adjusted to be 100 °C, 150 °C, 200 °C, 250 °C, and 300 °C. The DC magnetron sputtering method was used for the experiments. The distance between the target and the substrate was kept at 5 cm. The sputtering power was set at 60 W. The sputtering atmosphere pressure in the vacuum chamber was set at 2 Pa. The sputtering Ar/O_2_ ratio and flow rate were set at 60/100 sccm. The sputtering duration was 60 min. During the sputtering process, the substrate was fixed on a turntable in the sputtering chamber, which maintained a spin state at a rotation rate of 5 revolutions per minute (RPM). The resulting tungsten trioxide thin films were noted as WO_3_−RT, WO_3_−100, WO_3_−150, WO_3_−200, WO_3_−250, and WO_3_−300.

### 2.3. Characterizations

The crystallinity of WO_3_-based thin films coated onto the ITO glass was analyzed using X-ray diffraction (XRD, X’Pert PRO MPD, PANalytical, Almelo, The Netherlands) equipped with a CuKα (λ = 1.5418 Å) filter. Raman spectra were characterized by a micro confocal laser Raman spectrometer (Raman, LabRAM HR Evolution, HORIBA JOBIN YVON, Paris, France) with a laser wavelength of 532 nm. The structure of the sample was identified by field emission transmission electron microscopy (TEM, JEM-2100F, JEOL, Ltd., Tokyo, Japan) at an operating voltage of 200 kV. The morphologies of all samples were identified by high-resolution field emission scanning electron microscopy (HRSEM, Regulus 8230, Hitachi Ltd., Tokyo, Japan) at an operating voltage of 3 kV. The morphologies and roughness of all samples were identified by atomic force microscopy (AFM, Dimension Icon, Bruker, Saarbrücken, Germany) in an area of 2 × 2 μm^2^. The composition of samples was characterized by X-ray photoelectron spectroscopy (XPS, ESCALAB250Xi, Thermo, Waltham, MA, USA).

### 2.4. Electrochromic Performance Test

The electrochemical workstation (CHI760E, CH Instruments, Shanghai, China) was tested using a three-electrode system. The working electrode was the synthesized WO_3_ film. The reference electrode was Ag/AgCl. The counter electrode was Pt. One molar of LiClO_4_/PC solution was used as the electrolyte for performance testing. The electrochemical workstation was used in conjunction with a UV-VIS-NIR spectrophotometer (UV-3600, SHIMADZU, Tokyo, Japan) to measure the kinetic transition spectra and transmission spectra. The colored or bleached WO_3_ films were removed from the electrolyte to take the digital photos, dried, and then their CIE Lab chromaticity coordinates were measured with a spectrophotometer (WN700S, Shenzhen Weifu Optoelectronic Technology Co., Ltd., Shenzhen, China).

## 3. Results and Discussion

### 3.1. Structural Analysis

Figure 2a presents the X-ray diffraction (XRD) patterns of the tungsten trioxide thin films. It is evident that the samples WO_3_−RT, WO_3_−100, WO_3_−150, and WO_3_−200 did not exhibit discernible diffraction peaks, indicating that these films remained in an amorphous state when the in situ heating temperature did not exceed 200 °C [9]. However, in the case of WO_3_−250, several distinct diffraction peaks became apparent. These peaks can be indexed to specific crystal planes when compared to the standard diffraction pattern (PDF#43-1035), as detailed in Table 1. The observed diffraction angles, 2θ, and their corresponding crystal planes were as follows: 23.119° corresponds to the (002) plane, 23.586° corresponds to the (020) plane, 24.380° corresponds to the (200) plane, and 33.266° corresponds to the (022) plane. Among these peaks, the diffraction peak at the (002) crystal plane exhibited the highest intensity, suggesting a preferential orientation along this plane in the WO_3_−250 film in situ heating. The orientation of the crystal surface of (002) can improve the electrochemical activity of WO_3_ [10,11]. The diffraction intensity of each crystallographic plane for WO_3_−300 was found to be further enhanced, indicating an increase in crystallinity. This enhancement can be attributed to the diffusion of both tungsten (W) and oxygen (O) atoms as they were deposited onto the substrate. The diffusion ability of these atoms is governed by a set of Equations (2)–(5), which collectively describe the atomic mobility and the kinetics of diffusion during the in situ heating process.
(2)$$\tau_{a}=\tau_{0}exp(E_{d}/kT)$$
(3)$$\tau_{D}=\tau_{0}^{\prime }exp(E_{D}/kT)$$
(4)$$D_{s}=a_{0}^{2}∕\tau_{D}$$
(5)$$\bar{x}={\left(D_{s}\cdot \tau_{a}\right)}^{\frac{1}{2}}=a_{0}exp\left[\left(E_{d}-E_{D}\right)∕2kT\right]$$
where $\tau_{a}$ is the average adsorption time, $\tau_{0}$ is the adsorption time constant, $E_{d}$ is the desorption activation energy for chemisorption, *k* is the Boltzmann constant, *T* is the substrate surface temperature, $\tau_{D}$ is the average surface diffusion time, $\tau_{0}^{\prime }$ is the period of the atom’s vibration in the horizontal direction along the surface, $E_{D}$ is the surface diffusion activation energy, $\bar{x}$ is the average surface diffusion distance, $D_{s}$ is the surface diffusion coefficient, $a_{0}$ is the interval between neighboring adsorption positions. According to Equations (2)–(5), with increasing the in situ heating temperature, the average diffusion time ($\tau_{D}$) and the average surface diffusion distance ($\bar{x}$) of W and O atoms in the WO_3_ films increased. The enhanced diffusion capability enabled the W and O atoms to occupy their lattice positions through diffusion [12]. When compared to WO_3_−250, WO_3_−300 exhibited narrower half-peak widths in the diffraction peaks, suggesting that the WO_3_ grains refined further at the higher heating temperature. This refinement is indicative of improved crystallinity and more ordered atomic arrangements within the film.

Figure 2d displays the Raman spectra of the WO_3_ thin films. The spectrum revealed a broad and weak vibrational peak in the range of 515 to 880 cm^−1^ for the samples WO_3_−RT, WO_3_−100, WO_3_−150, and WO_3_−200. This peak corresponds to the vibrational mode of W^6+^-O bonding [13], indicative of the amorphous phase of WO_3_, in concordance with the XRD findings. However, in the spectra of WO_3_−250 and WO_3_−300, this broad peak split into two distinct peaks at 688 cm^−1^ and 790 cm^−1^ within the same spectral range. These peaks are attributed to the O-W-O stretching modes in the crystalline phase of WO_3_ [13], further confirming the crystallinity of WO_3_−250 and WO_3_−300 in alignment with the XRD results.

The vibrational peak observed at 260 cm^−1^ in the WO_3_ films could not be definitively attributed due to the confounding influence of the ITO substrate. It remains unclear whether this peak was associated with the W^4+^-O bonding or was an artifact of the ITO substrate [14]. Additionally, the peak at 950 cm^−1^ corresponded to the W^6+^=O terminal stretching vibration, which is induced by the absorption of water molecules by WO_3_ [15,16].

Figure 2b,c presents the transmission electron microscope (TEM) images of the WO_3_−250 sample after it was scraped off with a razor blade. These images revealed that the WO_3_−250 film was relatively thick when observed under TEM [17]. High-resolution TEM (HRTEM) images, shown in Figure 2e,f, were obtained by magnifying the areas where the film edges were thinner. These images allowed for the detection of lattice spacings corresponding to the crystal planes (002), (020), (200), and (022). The HRTEM results were in agreement with the XRD characterization, confirming the presence of these crystal planes in the WO_3_−250 film.

### 3.2. Surface Morphology Analysis

The HRSEM of the surface morphology and cross-section of WO_3_ thin films at different in situ heating temperatures are shown in Figure 3. It can be seen that the surface morphology of the films became less compact with increasing cracks as the in situ heating temperature increased [18].

Sputter deposition coating was also a nucleation and growth process, and the nucleation rate *N* of the new phase can be expressed by Equation (6):(6)$$N=Cexp\left(-A/kT\right)exp\left(-Q/kT\right)$$
where *C* is a constant. *A* is the nucleation work factor. *Q* is the atomic diffusion chance factor. *k* is the Boltzmann constant, and *T* is the substrate surface temperature. So, with the increase in situ heating temperature, the nucleation rate (*N*) increased, which promoted the nucleation of WO_3_, which led to more WO_3_ nuclei competing for growth, therefore leading to a reduction in grain size. We measured about 50 grain sizes per sample based on SEM images. The grain size distribution of WO_3_ samples is shown in Figure 4. Comparing the grain size distribution of the samples, we are informed that the grain size decreased with an increase in in situ heating temperature.

With the introduction of the in situ heating process and the subsequent increase in in situ heating temperature, the thickness of the WO_3_ films was observed to increase. HRSEM cross-sections revealed that the thicknesses of the films for WO_3_−RT, WO_3_−100, WO_3_−150, WO_3_−200, WO_3_−250, and WO_3_−300 were approximately 280 nm, 350 nm, 360 nm, 360 nm, 370 nm, and 370 nm, respectively. This increase in film thickness can be attributed to two primary factors.

First, the in situ heating process is suspected to have converted the physically adsorbed W and O atoms on the substrate into chemisorbed species [19], substantially increasing the amount of adsorbed material. This was reflected in the WO_3_−RT film, which had a thickness of only about 280 nm, whereas the films for WO_3_−100, WO_3_−150, WO_3_−200, WO_3_−250, and WO_3_−300 exhibited significantly greater thicknesses, ranging from 350 nm to 370 nm.

Second, a higher in situ heating temperature conferred greater diffusion ability to the W and O atoms [8,20]. This enhanced diffusion allowed for a more regular rearrangement of atoms from the disordered amorphous WO_3_ structure to a crystalline formation, leading to a slight increase in the thicknesses of the films for the WO_3_−100, WO_3_−150, WO_3_−200, WO_3_−250, and WO_3_−300 samples.

In order to scrutinize the surface morphology and roughness of the WO_3_ films, AFM tests were conducted. The roughness of each film is shown in Table 2. The corresponding root-mean-square (RMS) roughness (Rq) for WO_3_−RT, WO_3_−100, WO_3_−150, WO_3_−200, WO_3_−250, and WO_3_−300 were 2.57 nm, 2.77 nm, 2.96 nm, 3.33 nm, 3.55 nm, and 3.95 nm, respectively. The arithmetic mean roughness (Ra) was 2.04 nm, 2.22 nm, 2.33 nm, 2.64 nm, 2.82 nm, 3.15 nm. The surface roughness of the WO_3_ films was found to increase correspondingly with the rise in in situ heating temperature. Atomic Force Microscopy (AFM) images were shown in Figure 5. It revealed that the surface of the WO_3_-RT films predominantly featured the growth of smaller grains, which were stacked and distributed relatively uniformly across the surface [21]. As the in situ heating temperature was applied and increased, the film’s thickness augmented, and the grains grew at a faster rate, resulting in a rougher surface topology. This change in surface morphology is indicative of the altered growth dynamics and structural rearrangements occurring within the film as a function of the in situ heating temperature. After applying the in situ heating process, during the grain growth process, the surface morphology of the heated samples showed the phenomenon of unequal grain size and uneven distribution. The observed increase in surface roughness can be attributed to the higher surface energy associated with smaller grains. To minimize this surface energy, a process of grain coalescence, known as Osvaldo annexation or fusion, may occur. This process involves the merging of smaller grains to form larger grains [22,23], which in turn leads to an increase in surface roughness. The enhanced roughness was further compounded by the appearance of more cracks in the surface morphology, as evidenced in the HRSEM images. However, in the process of grain coalescence, the small grains do not form a complete large grain. We can be informed from Figure 3 that HRSEM images showed more and more cracks and smaller and smaller grain sizes when the in situ heating temperature increased. Meanwhile, these cracks were also a consequence of the stress buildup and relief during the grain growth and coalescence process, which can be exacerbated by the increasing roughness of the film surface.

### 3.3. Chemical Valence Analysis

To characterize the compositional differences and chemical valence state in the synthesized WO_3_ thin films, XPS tests were conducted, yielding a series of XPS spectra as presented in Figure 6. Figure 6a reveals a significant decrease in the binding energy of the W 4f peak for WO_3_ beginning at the in situ heating temperature of 200 °C. A similar trend was observed for the O 1s peak in Figure 6b, indicating that increasing the in situ heating temperature enhanced the bonding between W and O atoms, thereby favoring the formation of crystalline structures. The atomic proportions of W and O elements in the prepared WO_3_ samples are listed in Table 3, showing stable ratios without substantial variation.

The W 4f spectra, shown in Figure 6c–h, revealed the presence of both W^6+^ and a minor amount of W^5+^ valence states [24]. By calculating the area of the W4f spectral peaks for W^6+^ and W^5+^, the percentage of W^5+^ in the elemental W content was determined. The W^5+^ contents in WO_3_−RT, WO_3_−100, WO_3_−150, WO_3_−200, WO_3_−250, and WO_3_−300 were found to be approximately 0.92%, 0.92%, 1.08%, 1.24%, 1.58%, and 2.50%, respectively, indicating an increase with the increase in in situ heating temperature. This increase in W^5+^ content provides more active sites for Li ions and electrons to intercalate and de-intercalate during the electrochromic reaction [25].

The O 1s spectra In Figure 6i–n exhibited three oxygen states: W^6+^-O bonding from 530 to 531 eV, W^5+^-O bonding from 531 to 532 eV, and adsorbed water on the surface from 532 to 533 eV. The percentages of W^5+^-O bonding in WO_3_−RT, WO_3_−100, WO_3_−150, WO_3_−200, WO_3_−250, and WO_3_−300 were approximately 20.49%, 17.97%, 24.79%, 24.77%, 23.54%, and 24.38%, respectively, showing an overall slight increasing trend that corresponds to the rise in W^5+^ content. In the process of sputter coating, some point defects inevitably occur in the atomic arrangement, especially vacancies. In the O 1s spectra, O had a state that was bonded to W^5+^. However, in the W 4f spectra, there might be a possible dangling bond of the W ion due to the presence of oxygen vacancies. This may account for the difference in W^5+^ content between W 4f and O 1s spectra.

### 3.4. Electrochromic Performance Testing and Analysis

Figure 7a,d illustrates that the window voltages and peak potentials observed during the electrochemical testing of the WO_3_ thin films expanded as the in situ heating temperature was elevated. This expansion can be attributed to the previously described crystallization of the WO_3_ structure that occurred with increasing in situ heating temperature. The crystalline structure necessitated a higher voltage to facilitate the intercalation of Li ions and electrons within the WO_3_ film. Concurrently, the higher in situ heating temperature resulted in thicker films with a greater volume of WO_3_ material, which in turn accommodated a larger number of Li ions and electrons, thereby increasing the CV closure area. Consequently, unless otherwise specified in the subsequent sections, the window voltage was set to −1.0 to +1.0 V for the WO_3_−RT, WO_3_−100, WO_3_−150, and WO_3_−200 samples, and −1.0 to +2.0 V for the WO_3_−250 and WO_3_−300 samples.

The kinetic transition spectra of different samples with applied −1.0 V and +1.0 V potentials, −1.0 V and +2.0 V potentials, and their partial magnification were shown in Figure 7b,c,e, and f, respectively. And their corresponding coloring and bleaching times were listed in Table 4.

Figure 7b,e demonstrates that as the in situ heating temperature increased, the WO_3_ structure underwent crystallization when subjected to −1.0 V and +1.0 V potentials. The +1.0 V potential significantly hindered the disembedding extraction of Li ions and electrons, effectively trapping them within the lattice and creating potential traps [26]. This impeded the bleaching rate, prolonging the bleaching time from 13.4 s for WO_3_-RT to 33.6 s for WO_3_−250. In the case of WO_3_−300, the color could not be fully bleached even within 60 s. The coloring times for all samples, including WO_3_−300, stabilized within the range of 20 to 25 s. However, for the four amorphous samples—WO_3_−RT, WO_3_−100, WO_3_−150, and WO_3_−200—the coloring rate initially slowed before accelerating. This behavior may be attributed to the initial hindrance of ion and electron embedding due to crystallization [27], followed by an enhanced coloring rate as a result of the roughened surface morphology, which provides a larger specific surface area and multiple embedding sites.

In Figure 7c,f, the coloring potential was set to −1.0 V, and the bleaching potential was adjusted to +2.0 V, in accordance with the expanded window voltage, as the +1.0 V potential was insufficient to completely bleach WO_3_−250 and WO_3_−300. At this new kinetic transition potential, the kinetic transition performance of WO_3_−250 showed significant improvement compared to the −1.0 V and 1.0 V conditions. The coloring time remained largely consistent, with a slight improvement from 20.2 s to 20.0 s, while the bleaching time was reduced from 33.6 s to 19.4 s, substantially enhancing the bleaching rate. For WO_3_−300, the coloring time of 20.8 s was reduced to a successful bleaching time of 41.2 s, marking a considerable improvement from the previous challenge of achieving complete bleaching [28].

The transmission spectra and digital photographs depicting the various states of WO_3_ films with different parameters are presented in Figure 8. The color of the films is represented using CIE Lab color space coordinates. The L* value, ranging from 0 to 100, indicates the lightness or darkness of the film, with lower values corresponding to darker shades and higher values to lighter shades. The a* value, ranging from −128 to 127, signifies the redness or greenness of the film, with positive values indicating a shift toward red and negative values toward green. The b* value, also ranging from 128 to 127, represents the yellowness or blueness of the film, with positive values indicating a shift towards yellow and negative values towards blue. The transmittance and chromaticity coordinates of the WO_3_ films in the colored and bleached states are listed in Table 5.

As the in situ heating temperature increased, the surface roughness of the film surface also increased, leading to enhanced surface scattering [28,29]. This scattering effect caused a downward shift in the transmission spectrum during the colored state, with the transmittance at 633 nm decreasing from 24.8% for WO_3_−RT to 16.0% for WO_3_−300. Additionally, the L* value in the CIE Lab chromaticity coordinates decreased, indicating that the color of the film became progressively darker in terms of lightness and darkness [30,31]. This suggests that the films exhibited a higher level of coloration with an increase in in situ heating temperature, which could be beneficial for applications requiring deep, saturated colors.

The L-value of the colored state decreased from 46.41 to 29.32 (min. in WO_3_−250) with an increase in in situ heating temperature, resulting in a darker color. Upon examination of the coloring state diagram for the WO_3_ film, it is observed that the blue hue underwent a transition from a gray-blue to a darker blue as the in situ heating temperature elevated. The b* value, which characterizes the blueness or yellowness of the film, continuously decreased from −1.23 for WO_3_−RT to −24.06 for WO_3_−300, signifying a pronounced intensification of the blue color. This shift is attributed to the increase in the W^5+^ content within the films that resulted from the elevated in situ heating temperature. The presence of W^5+^ is known to influence the optical properties of WO_3_, particularly its coloration behavior, leading to the observed deepening of the blue color with increasing in situ heating temperature. The increase in the initial W^5+^ content provided more reaction sites, which benefited the embedding of Li ions. The (200) lattice plane orientation had high electrochemical activity [32], which benefited the color change reaction. During the electrochromic reaction, more W^6+^ was converted to W^5+^, resulting in a bluer color. As a result, the color of WO_3_ changed from gray-blue to dark-blue due to roughness-enhanced surface scattering and changes in the valence state of the W element. For the bleached state of the samples, the range of WO_3_ films maintained their inherent advantages of colorless transparency and high transmittance, with only minimal overall changes [24,33]. Consequently, the contrast at 633 nm demonstrated an overall increasing trend. In the amorphous WO_3_, the transmittance continued to increase with a maximum value of 81.3% at 200 °C, which slightly decreased to 77.7% as the temperature rose to 250 °C, marking the transition to crystalline WO_3_. Subsequently, the transmittance increased again to 80.7% as the in situ heating temperature was further increased to 300 °C. Across all the samples, WO_3_−250 exhibited the darkest colored state and the highest contrast at 633 nm, offering a superior range of optical modulation and color change.

The coloring efficiency (*CE*) is also a very important indicator of electrochromic performance. *CE* is the ratio of the change in optical density caused by the amount of injected charge per unit area, which is calculated by Equations (7) and (8): (7)$$\Delta OD=\log\frac{T_{b}}{T_{c}}$$
(8)$$CE=\frac{\Delta OD}{Q}$$
where Δ*OD* represents the change in optical density, and *Q* is the amount of charge injected per unit area. The coloring efficiency of WO_3_ films is shown in Figure 9. In the amorphous WO_3_ at temperatures below 200 °C, the coloring efficiency (*CE*) of the thin-film samples displayed a pattern of increasing and then slightly decreasing with the increase in in situ heating temperature. The WO_3_−150 sample reached a maximum *CE* of 39.3 cm^2^/C, attributed to the in situ heating temperature’s effect on reducing the transmittance of the colored state, thereby increasing the optical density change in the film. The crystallization of WO_3_ occurs in the temperature range of 200 to 250 °C [34,35]. The *CE* of WO_3_−250 slightly decreased compared to WO_3_−200, and the *CE* of WO_3_−300 further decreased to 35.2 cm^2^/C. At this temperature, while the transmittance of the colored state continued to decrease [35], the capacity for accommodating Li ions and electrons increased due to the more stable crystalline structure compared to the amorphous state and the presence of more oxygen vacancies, which led to an increase in the capacity for Li ions and electrons [36,37]. This increased capacity for Li ions and electrons also resulted in a decrease in the coloring efficiency of the crystalline WO_3_ films.

### 3.5. Cyclic Stability Testing and Analysis

In order to compare the cycling stability between amorphous and crystalline WO_3_, we performed kinetic conversion tests on samples WO_3_−RT and WO_3_−250 for 1500 and 6000 cycles, respectively. As shown in Figure 10, sample WO_3_-RT showed a large change in current density during the cycling test. This was due to the fact that its amorphous structure could no longer maintain structural stability during the long-term electrochemical reaction. The electrolyte for the sample WO_3_−250 was refilled after 3000 cycles, which showed a sharp change in the measurement plot. Compared to WO_3_−RT, no evident decline was observed in the current density.

During the 1500-cycle testing of the WO_3_−RT sample, transmission spectra of both the colored and bleached states were measured at regular intervals of 500 cycles, and digital photographs of the colored and bleached states were taken to document the changes over the cycling process. These measured values are compiled in Table 6. As observed in Figure 11, the stability of the WO_3_−RT sample was not satisfactory. The film exhibited partial damage and lost its ability to change color after only 500 cycles, with the damage becoming more pronounced as the number of cycles increased. The electrochromic performance and chromaticity at the center of the film were measured and compared to the initial state, which exhibited 70.1% contrast at 633 nm. After 1000 cycles, the contrast was reduced to 71.7% at 633 nm, indicating a less severe depletion in electrochromic performance. However, after 1500 cycles, the overall structural damage to the film resulted in a significant degradation of the electrochromic performance, leading to the loss of the color change function. WO_3_−RT was an amorphous structure characterized by short-range disordered atomic arrangements, which were unable to withstand the continuous intercalation and deintercalation of ions and electrons during prolonged cycling tests.

Throughout the 6000-cycle durability testing of the WO_3_−250 sample, transmission spectra of both the colored and bleached states were measured every 1000 cycles, and digital photographs of the colored and bleached states were captured to monitor the changes throughout the cycling process. The measured values are tabulated in Table 7. Figure 12 reveals that as the number of cycles increased, the colored state of WO_3_−250 shifted toward a bluer hue. This trend was quantified by measuring the CIE Lab chromaticity spatial coordinates, which showed a clear decrease in the b* value, indicating a bluer color. This shift may be attributed to the increase in W^5+^ content during cycling, as some Li ions were trapped by the WO_3_ lattice to form potential wells, inhibiting their release. This also led to a decrease in the transmittance of the film in the bleached state and a reduction in the optical modulation range.

For the first 5000 cycles, the L-value remained relatively constant, at around 30, suggesting that the depth of the color did not significantly change. At this point, the contrast at 633 nm was 68.4%, which retained 88.0% of the original performance. However, upon cycling to 6000 cycles, the electrochromic performance experienced a significant degradation to 60.2%, representing 77.5% of the initial performance. The crystalline structure of WO_3_−250, coupled with the excellent electrochemical activity resulting from the (002) selective orientation, contributed to a remarkable cycle life of 5000 cycles. This structural integrity and electrochemical activity ensured that there was no significant degradation in performance over the initial 5000 cycles [38,39].

## 4. Conclusions

In this study, we fabricated a series of WO_3_ thin film samples by modulating the in situ heating process. The crystalline transition temperature was observed to be approximately 200 to 250 °C, resulting in the formation of a (002) lattice plane selective orientation. This selective orientation endowed crystalline WO_3_ with superior electrochemical activity and cycling stability. As the in situ heating temperature increased, the films exhibited grain refinement during growth, accompanied by the processes of grain coalescence or fusion. The increased roughness leads to enhanced surface scattering, which, in turn, reduced the transmittance and deepened the color in the colored state. Moreover, with the rise in the in situ heating temperature, the content of W^5+^ in WO_3_ increased, resulting in a deeper blue color in the colored state. The WO_3_ films had just completed the crystal structure transition at 250 °C. At this temperature, compared to 300 °C in situ heating, the hindrance to Li ion and electron delocalization was not yet pronounced, and the response rate remained relatively fast (tc = 20.0 s, tb = 19.4 s). When compared to the amorphous films, the crystalline structure demonstrated an excellent overall performance (ΔT = 77.7%, CE = 37.1 cm^2^/C, and L* = 29.32, a* = 7.41, b* = −22.12 for the colored state), particularly in terms of cyclic stability (5000 cycles). In summary, this work revealed the growth behaviors of tungsten oxide under in situ heating mode by reactive magnetron sputtering, as well as their effect on the electrochromic performance of the films. Our work may provide useful information for designing electrochromic materials and smart devices. 

## Figures and Tables

**Figure 1 materials-17-02214-f001:**
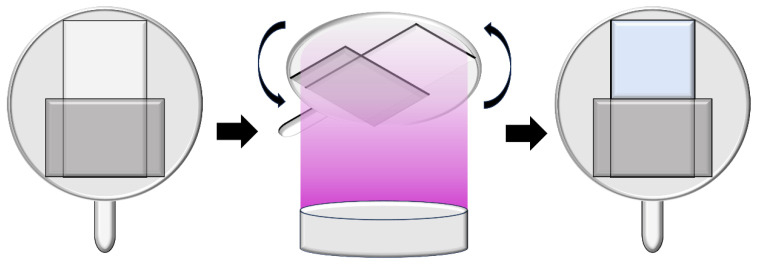
Schematic of magnetron sputtering coating.

**Figure 2 materials-17-02214-f002:**
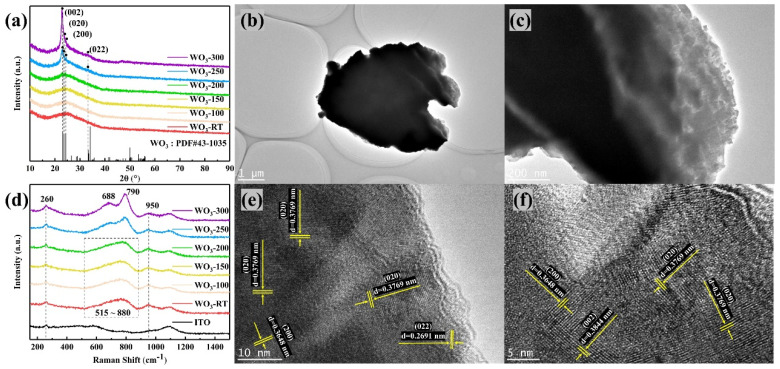
(**a**) XRD, (**d**) Raman analysis of six WO_3_ samples, and (**b**,**c**) TEM images, (**e**,**f**) HRTEM images of WO_3_−250.

**Figure 3 materials-17-02214-f003:**
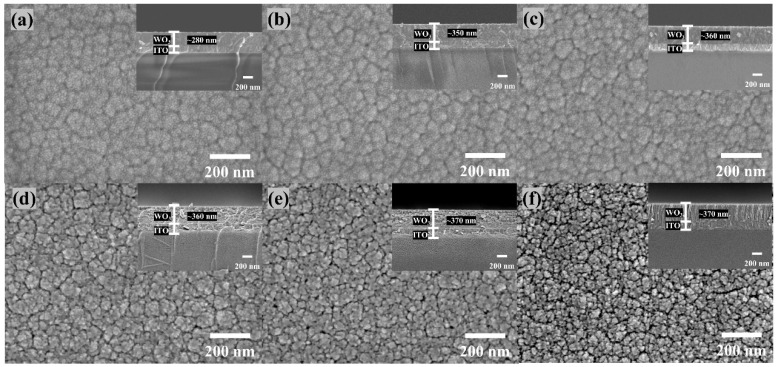
Surface morphology and cross-sectional HRSEM images of (**a**) WO_3_−RT, (**b**) WO_3_−100, (**c**) WO_3_−150, (**d**) WO_3_−200, (**e**) WO_3_−250, (**f**) WO_3_−300.

**Figure 4 materials-17-02214-f004:**
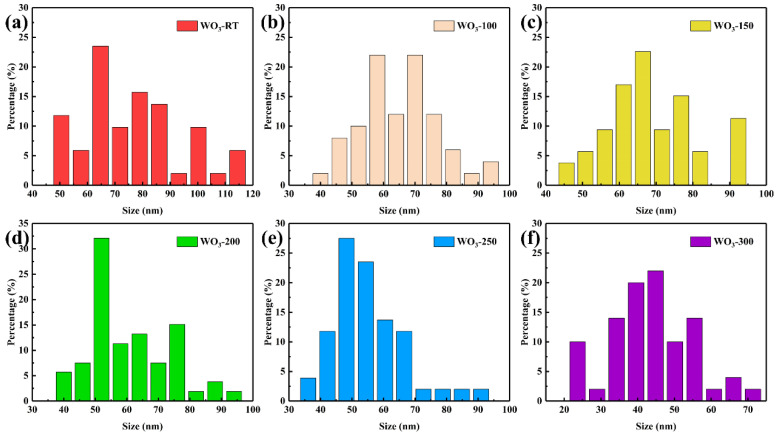
Grain sizes distribution diagram of (**a**) WO_3_−RT, (**b**) WO_3_−100, (**c**) WO_3_−150, (**d**) WO_3_−200, (**e**) WO_3_−250, (**f**) WO_3_−300.

**Figure 5 materials-17-02214-f005:**
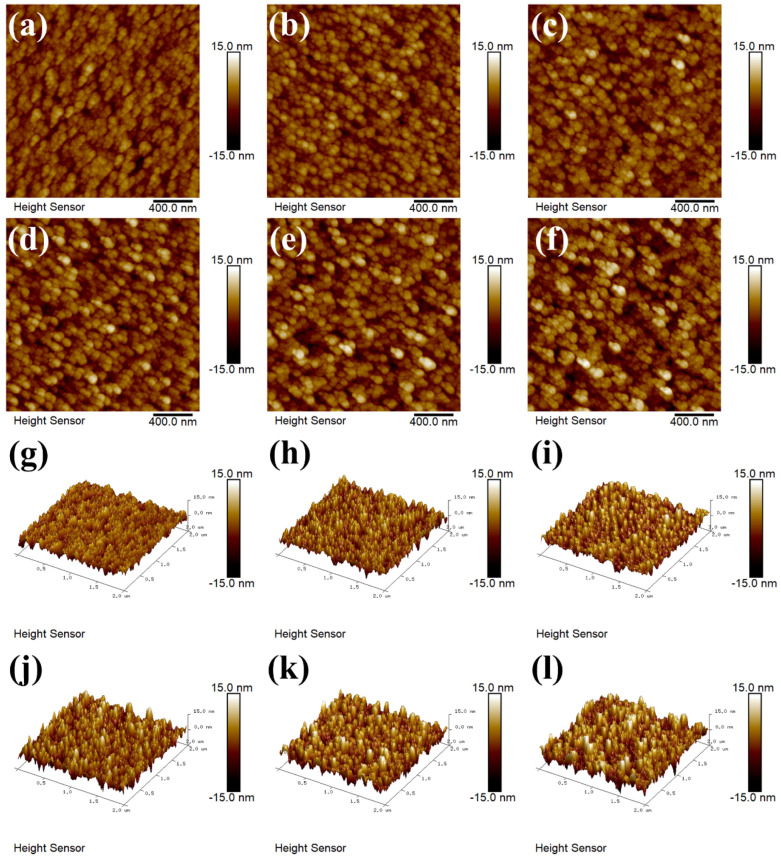
AFM 2D images of (**a**) WO_3_−RT, (**b**) WO_3_−100, (**c**) WO_3_−150, (**d**) WO_3_−200, (**e**) WO_3_−250, (**f**) WO_3_−300. AFM 3D images of (**g**) WO_3_−RT, (**h**) WO_3_−100, (**i**) WO_3_−150, (**j**) WO_3_−200, (**k**) WO_3_−250, (**l**) WO_3_−300.

**Figure 6 materials-17-02214-f006:**
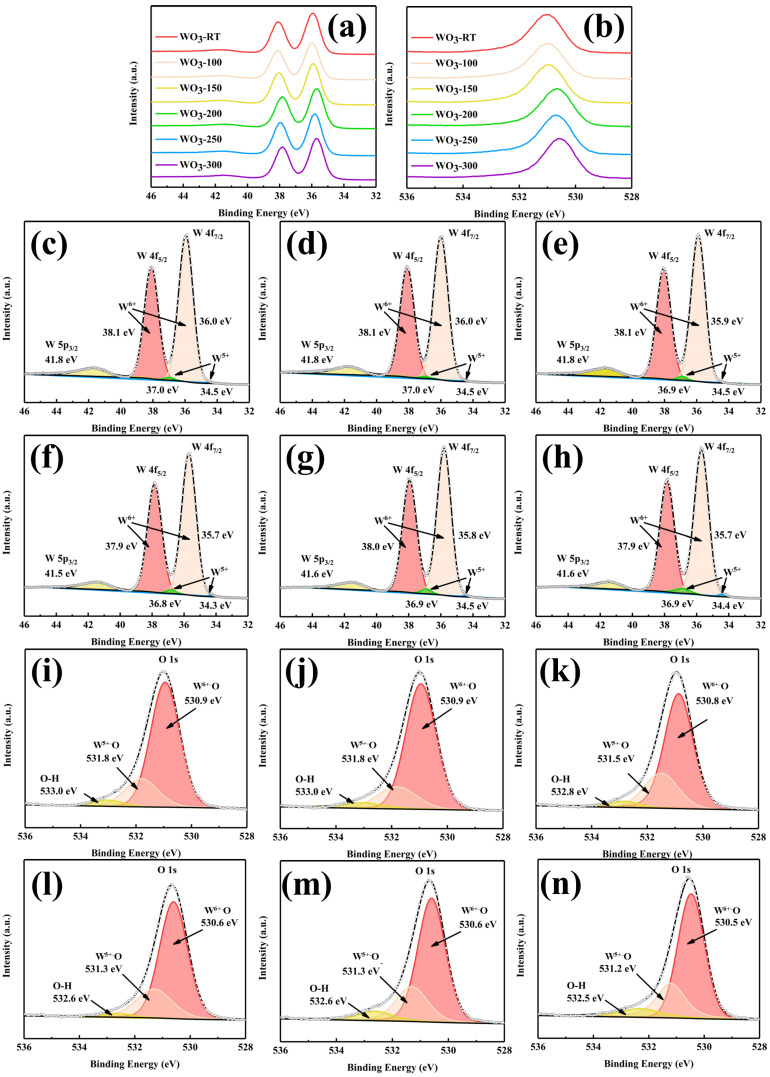
A series of (**a**) W 4f, (**b**) O 1s spectra of synthesized WO_3_, W 4f spectra of (**c**) WO_3_−RT, (**d**) WO_3_−100, (**e**) WO_3_−150, (**f**) WO_3_−200, (**g**) WO_3_−250, (**h**) WO_3_−300. O 1s spectra of (**i**) WO_3_−RT, (**j**) WO_3_−100, (**k**) WO_3_−150, (**l**) WO_3_−200, (**m**) WO_3_−250, (**n**) WO_3_−300.

**Figure 7 materials-17-02214-f007:**
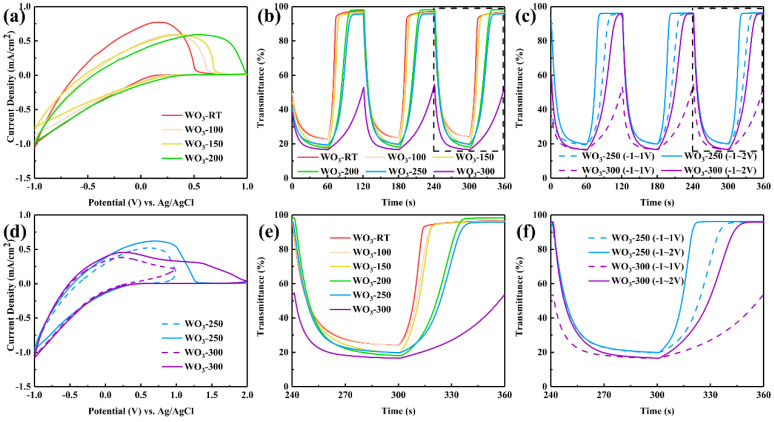
The cyclic voltammetry curves of a series of samples of window potential were (**a**) −1.0 V~+1.0 V and (**d**) −1.0 V~+2.0 V. The kinetic transition spectra of a series of samples with applied (**b**) −1.0 and +1.0 V potentials, (**c**) their partial magnification, (**e**) −1.0 V and +2.0 V potentials, and (**f**) their partial magnification.

**Figure 8 materials-17-02214-f008:**
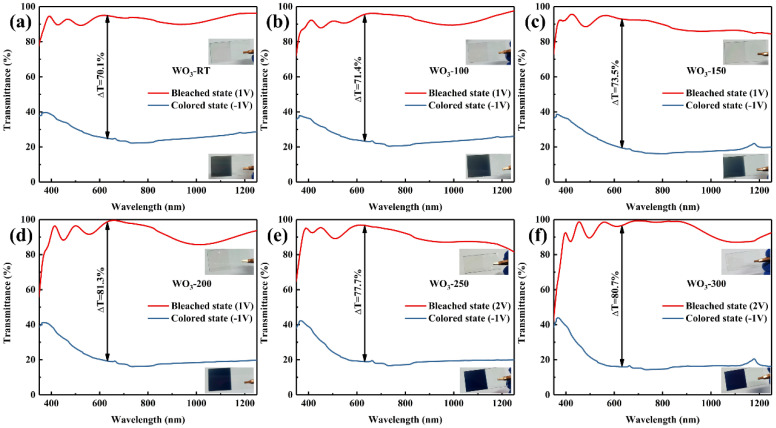
The transmission spectra along with digital photos in different states of (**a**) WO_3_−RT, (**b**) WO_3_−100, (**c**) WO_3_−150, (**d**) WO_3_−200, (**e**) WO_3_−250, (**f**) WO_3_−300.

**Figure 9 materials-17-02214-f009:**
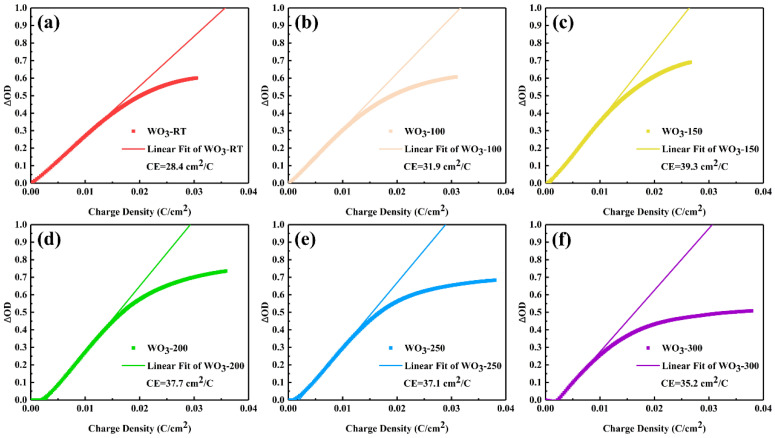
CE of a series of synthesized WO_3_ at 633 nm: (**a**) WO_3_−RT (**b**) WO_3_−100 (**c**) WO_3_−150 (**d**) WO_3_−200 (**e**) WO_3_−250 and (**f**) WO_3_−300.

**Figure 10 materials-17-02214-f010:**
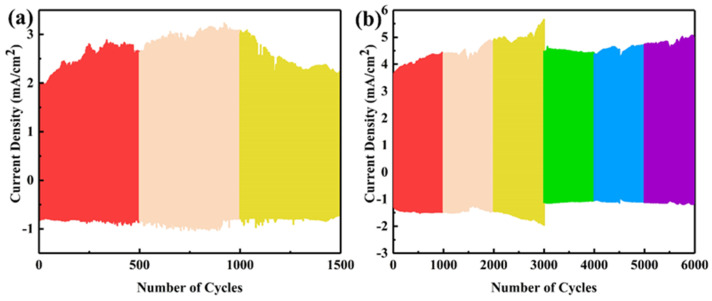
Kinetic conversion cycle current density plot of the samples (**a**) WO_3_-RT and (**b**) WO_3_−250.

**Figure 11 materials-17-02214-f011:**
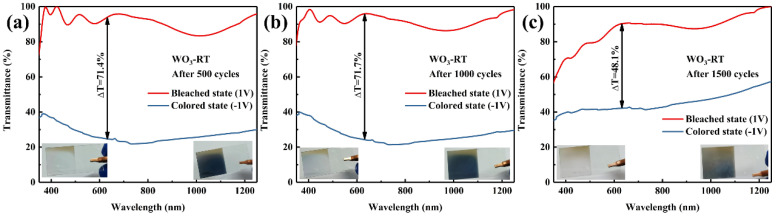
Transmission spectra of colored and bleached states of sample WO_3_−RT cycled (**a**) 500 times, (**b**) 1000 times, (**c**) 1500 times, and their physical, digital photographs.

**Figure 12 materials-17-02214-f012:**
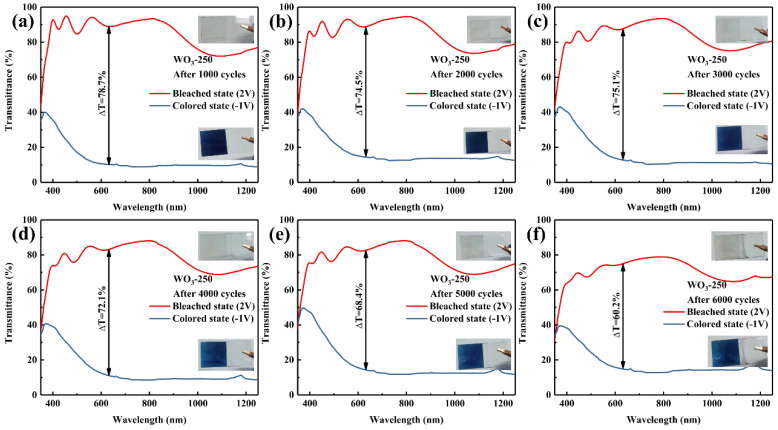
Transmission spectra of colored and bleached states of sample WO_3_−250 cycled (**a**) 1000 times, (**b**) 2000 times, (**c**) 3000 times, (**d**) 4000 times, (**e**) 5000 times, (**f**) 6000 times and their physical, digital photographs.

**Table 1 materials-17-02214-t001:** 2 Theta angle and corresponding crystal plane.

2 Theta (°)	Lattice Plane
23.119	(002)
23.586	(020)
24.380	(200)
33.266	(022)

**Table 2 materials-17-02214-t002:** Surface roughness of six samples.

Sample	WO_3_−RT	WO_3_−100	WO_3_−150	WO_3_−200	WO_3_−250	WO_3_−300
Rq/nm	2.57	2.77	2.96	3.33	3.55	3.95
Ra/nm	2.04	2.22	2.33	2.64	2.82	3.15

**Table 3 materials-17-02214-t003:** The proportion of W and O atoms in the samples.

Samples	W (at%)	O (at%)
WO_3_−RT	23.09	76.91
WO_3_−100	22.42	77.58
WO_3_−150	23.10	76.90
WO_3_−200	23.61	76.39
WO_3_−250	22.74	77.26
WO_3_−300	22.81	77.19

**Table 4 materials-17-02214-t004:** Kinetic response times of a series of WO_3_ thin films.

Samples	WO_3_−RT	WO_3_−100	WO_3_−150	WO_3_−200	WO_3_−250	WO_3_−300
t_c_ (−1~1 V)/s	21.8	23.2	24.2	21.6	20.2	/
t_b_ (−1~1 V)/s	13.4	16.6	17.2	31.2	33.6	/
t_c_ (−1~2 V)/s	/	/	/	/	20.0	20.8
t_b_ (−1~2 V)/s	/	/	/	/	19.4	41.2

**Table 5 materials-17-02214-t005:** Transmittance and CIE Lab chromaticity coordinates of a series of synthesized WO_3_ at 633 nm.

Sample	WO_3_−RT	WO_3_−100	WO_3_−150	WO_3_−200	WO_3_−250	WO_3_−300
Transmittance of bleached state/% (@633 nm)	94.9	95.7	92.9	98.5	96.7	96.7
Transmittance of colored state/% (@633 nm)	24.8	24.3	19.4	17.2	19.0	16.0
Contrast/% (@633 nm)	70.1	71.4	73.5	81.3	77.7	80.7
L* of bleached state	84.12	86.12	87.25	87.82	87.77	88.13
a* of bleached state	−4.50	−2.50	−1.19	−2.33	−0.81	−4.23
b* of bleached state	−5.07	−5.29	−5.89	−3.95	−2.35	−1.66
L* of colored state	46.41	47.30	40.92	35.74	29.32	31.62
a* of colored state	−8.78	−7.19	−2.62	1.07	7.41	9.22
b* of colored state	−1.23	−4.35	−11.97	−18.31	−22.12	−24.06

**Table 6 materials-17-02214-t006:** Transmittance of WO_3_−RT in colored and bleached states and corresponding chromaticity coordinates during 1500 cycles.

No. of Cycles	0	500	1000	1500
Transmittance of bleached state/% (@633 nm)	94.9	96.1	95.9	90.3
Transmittance of colored state/% (@633 nm)	24.8	24.7	24.2	42.2
Contrast/% (@633 nm)	70.1	71.4	71.7	48.1
L* of bleached state	84.12	81.72	75.66	82.1
a* of bleached state	−4.5	−5.12	−5.57	−4.24
b* of bleached state	−5.07	−8.66	−7.34	1.49
L* of colored state	46.41	39.48	46.05	50.7
a* of colored state	−8.78	2.02	−8.12	−6.43
b* of colored state	−1.23	−8.64	−8.53	4.19

**Table 7 materials-17-02214-t007:** Transmittance of WO_3_−250 in colored and bleached states and corresponding chromaticity coordinates during 6000 cycles.

No. of Cycles	0	1000	2000	3000	4000	5000	6000
Transmittance of bleached state/% (@633 nm)	96.7	88.9	88.9	87.8	83.1	82.6	75.0
Transmittance of colored state/% (@633 nm)	19.0	10.2	14.4	12.7	11.0	14.2	14.8
Contrast/% (@633 nm)	77.7	78.7	74.5	75.1	72.1	68.4	60.2
L* of bleached state	87.77	88.21	86.27	82.19	81.36	82.43	74.11
a* of bleached state	−0.81	−3.33	−1.11	−0.72	0.25	−0.55	1.41
b* of bleached state	−2.35	−4.26	−1.27	−0.79	1.10	0.39	3.37
L* of colored state	29.32	27.54	34.34	28.74	27.64	31.95	41.33
a* of colored state	7.41	11.74	1.22	4.81	0.78	0.52	−7.20
b* of colored state	−22.12	−20.49	−29.00	−28.09	−26.57	−28.33	−21.33

## Data Availability

The data will be provided as required.

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
