# Peer review of "Effect of In Situ Heating on the Growth and Electrochromic Properties of Tungsten Trioxide Thin Films"

_materials, 2024, doi:10.3390/ma17102214_

Round 1
Reviewer 1 Report
Comments and Suggestions for Authors
The paper presents the effect of annealing on the WO3 oxide deposited by magnetron sputtering. The topic is not new, but the large quantity of analysis and results presented deserves to be commented on. The paper has significant and unacceptable problems; the scientific soundness is very low, most of the discussions are confusing, and there are wrong conclusions. The language's low quality does not help interpret the Authors messages. This work can’t be published in this form. Several major issues must be addressed, in more detail:
1. Abstract: English quality is deficient. The scientific case is not presented; only detailed conclusions are shown.
2. Introduction: the text in lines 31-36 must be reduced. Equation A1 must be explained. Lines 44-49: the Authors must properly introduce lithium's presence and charge transport's role.
3. Materials: line 61 and similar, mm should be cubic mm. Line 64: the meaning of backup substrate is unknown. Line 74: air pressure should be atmosphere pressure in the vacuum chamber. The inverted T shape subtrates are rotated (“self-rotation” is misleading) during deposition; the frequency is missing, and the Authors must comment on the stability of this setup, did they fixed the substrates? Line 83: 10-90 degrees? Pre-annealing: it is unclear if there was a final annealing and if the pre-annealing was done during the film deposition. The Authors should use the “annealing” term, in my opinion.
4. Results 3.1: the equations are useless; the Authors don’t discuss them and use general claims (lines 124-130).
5. Results 3.2: line 155: Fig.2 is Fig.3. Line 173: the WO3-RT thickness “only about 280nm”, did the Authors expected a higher value? Lines 176-177: this claim is unreadable. Lines 169-180: the Authors described two well-known and established phenomena, i.e., chemisorption and diffusion; they must reduce the length of their comments. Fig.3: these images require further comments, particularly about the presence of grains and columns in the samples, which do not entirely correspond to the XRD and Raman results. Lines 198-200: “leaded” is incorrect. Do the Authors really see differences in the samples for the grain size from SEM and AFM analysis? Apparently, this is not true; only a height difference is evident. To support their claims, the Authors must measure the grain mean dimensions for all samples.
6. Results 3.3: “split-peak fitting”, “XPS images” are unknown and wrong terminology. The binding energy “decreasing trend” with increasing pre-annealing temperature does not indicate W-O chemical bond formation. The core level BEs haven’t the tendency to decrease, they decrease. The spin-orbit splitting has a fixed value. Thus, the description of both doublet peaks is unnecessary, annoying, and misleading, as it suggests a variation of this splitting from 2.1 to 2.2 eV (the correct value is 2.2eV). Fig.5: “Ovacancy” label form must be changed. W5+: the presence of these compounds is very strange and unreliable. As an unstable oxide (no references are shown), it is probably related to a wrong peak fitting (remove the W5+ discussion in lines 297-299). The reported values (line 228) have an unreal precision. The oxygen vacancies, by definition, are missing oxygen atoms and thus not detectable (no references are shown); probably, these are interstitial atoms. The Authors did not show the W/O ratio for the WO3 compound, and did not comment on the BEs of W4f 7/2 and O1s, are these the proper values?
7. Results 3.4: Line 278: Fig.6 should be Fig. 7. Lines 255-256: how can the Authors detect a crystallization effect from Fig.5? The claims in lines 260-275 are confusing and unclear. L*, a*, b* parameters require a scientific and adequate explanation, and the range for the colors parameters is from -127 to +128. The CE results are shown in Fig.8, the Authors must refer to this figure.
8. Conclusions: lines 333-336 are repetitive and misleading. The Authors move from structural properties to voltammetry analysis, then to surface morphology and coloring efficiency, then to material composition, and so on. This section is highly confusing.
9. References: they are embedded and mixed in the text, it is often difficult to identify them. References numbering must be checked (e.g., see line 55).
10. English quality: very long sentences, or short sentences always starting with “And”. Several grammatical errors (e.g., leaded). Several typos are present. WO3 hasn’t the number as a subscript, very often.
Comments on the Quality of English LanguageThe English quality is very low: long sentences, or short sentences always starting with “And”. Several grammatical errors (e.g., leaded). Several typos are present. WO3 hasn’t the number as a subscript very often.
Author Response
We really appreciate the Editor and Reviewers’ constructive suggestions. We have made every effort to revise the manuscript following the Editor’s and Reviewers’ comments and suggestions. In this revision, we have rewritten a lot of parts. We really appreciate your understanding and support. The modifications have been highlighted in the revised manuscript.

Reviewer 2 Report
Comments and Suggestions for Authors
Please find the information in the attached file.

The English requires further improvement.
Author Response

(The authors gave the same response as above.)

Round 2
Reviewer 1 Report
Comments and Suggestions for Authors
The Authors have made significant strides in improving the English quality and scientific soundness of the paper, resulting in much improved readability. The discussion of results, especially the electrochromic data analysis, has been enhanced by adding new text and figures. The Authors must still address a few minor issues, in more detail:
1. Abstract: the Authors must reduce the technical details and summarize the overall electrochromic performance
2. Results: the statistical analysis in Fig.4 could be more representative, probably due to the low number of analyzed grains (50). Moreover, grain dimension distribution from HRSEM shows a mean size reduction with increasing annealing temperature. However, from AFM, a coalescence process is proposed to justify the roughness increase with annealing temperature; the Authors must comment on this apparent contradiction. Lines 195-196: change “chemisorption” with “chemisorbed species” to improve clarity. Fig.3: I suggest increasing the inset dimensions to improve readability. Presence of W5+ peaks: compared with the results of Ref.24, the presence of this new doublet could be reasonable. However, the Authors identified a component in the O1s core level related to oxygen atoms involved in W5+/species. Still, the weight of this component is about five times lower than oxygen in W6+/O species, while from W4f core level the W6+/W5+ ratio is at about twenty. The Authors must comment on this discrepancy. The precision used for the peaks relative weight is too high, i.e., 0.918% should be 0.92%.
3. Typos, errors: line 77 (And), 95 (the), 149 (2 in bold), 177 (increasing), 244 (at a), and others.
Comments on the Quality of English LanguageThe Authors highly improved English language, reaching the required quality for a scientific paper.
Author Response
We really appreciate the Editor and Reviewers’ constructive suggestions. We have made every effort to revise the manuscript following the Editor’s and Reviewers’ comments and suggestions. In this revision, we have corrected some errors. We really appreciate your understanding and support. The modifications have been highlighted in the revised manuscript.

Reviewer 2 Report
Comments and Suggestions for Authors
1. I am still not convinced by the W5+ fitted peak. Referring to comment 15 in the previous review, the author claims that removing the fitting peaks of W5+ would significantly increase the deviation between the fitting results and the XPS results. Could you provide the numerical comparison between the scenarios with and without W5+? This data does not need to be included in the main paper, but is required in response to the comment here.
2. In the manuscript's 'Cyclic Stability Testing and Analysis' section, particularly Figure 10 which shows the kinetic conversion cycle current density, why does the graph appear very fluctuating? Is it possible that your experimental setup was not stable? It should not behave like this. could you explain this?
Author Response

(The authors gave the same response as above.)
